# Inter-System Variability of Eight Different Handheld Ultrasound (HHUS) Devices—A Prospective Comparison of B-Scan Quality and Clinical Significance in Intensive Care

**DOI:** 10.3390/diagnostics14010054

**Published:** 2023-12-26

**Authors:** Johannes Matthias Weimer, Diana Beer, Christoph Schneider, Masuod Yousefzada, Michael Gottwald, Tim Felix Züllich, Andreas Weimer, Christopher Jonck, Holger Buggenhagen, Roman Kloeckner, Daniel Merkel

**Affiliations:** 1Rudolf Frey Learning Clinic, University Medical Center, Johannes Gutenberg University Mainz, 55131 Mainz, Germany; cjonck@uni-mainz.de (C.J.); buggenha@uni-mainz.de (H.B.); 2Immanuel Klinik Rüdersdorf, University Hospital of the Brandenburg Medical School, 15562 Rüdersdorf bei Berlin, Germany; diana.beer@immanuelalbertinen.de (D.B.); christoph.schneider@immanuelalbertinen.de (C.S.); masuod.yousefzada@immanuelalbertinen.de (M.Y.); michael.gottwald@immanuelalbertinen.de (M.G.); timfelix.zuellich@immanuelalbertinen.de (T.F.Z.); 3Center of Orthopedics, Trauma Surgery and Spinal Cord Injury, Heidelberg University Hospital, 69118 Heidelberg, Germany; andreas.weimer@kkh-bergstrasse.de; 4Institute of Interventional Radiology, University Hospital Schleswig-Holstein—Campus Lübeck, 23538 Lübeck, Germany; roman.kloeckner@uksh.de; 5BIKUS—Brandenburg Institute for Clinical Ultrasound, Brandenburg Medical School Theodor Fontane (MHB), 16816 Neuruppin, Germany

**Keywords:** B-scan, sonography, quality, comparison image quality, HEUS, high-end ultrasound, HHUS, handheld ultrasound, pocket ultrasound, intensive care

## Abstract

Background: the use of handheld ultrasonography (HHUS) devices is well established in prehospital emergency diagnostics, as well as in intensive care settings. This is based on several studies in which HHUS devices were compared to conventional high-end ultrasonography (HEUS) devices. Nonetheless, there is limited evidence regarding potential variations in B-scan quality among HHUS devices from various manufacturers, and regarding whether any such differences hold clinical significance in intensive care medicine settings. Methods: this study included the evaluation of eight HHUS devices sourced from diverse manufacturers. Ultrasound videos of five previously defined sonographic questions (volume status/inferior vena cava, pleural effusion, pulmonary B-lines, gallbladder, and needle tracking in situ) were recorded with all devices. The analogue recording of the same pathologies with a HEUS device served as gold standard. The corresponding findings (HHUS and HEUS) were then played side by side and evaluated by sixteen intensive care physicians experienced in sonography. The B-scan quality and the clinical significance of the HHUS were assessed using a five-point Likert scale (5 points = very good; 1 point = insufficient). Results: both in assessing the quality of B-scans and in their ability to answer clinical questions, the HHUS achieved convincing results—regardless of the manufacturer. For example, only 8.6% (B-scan quality) and 9.8% (clinical question) of all submitted assessments received an “insufficient” rating. One HHUS device showed a significantly higher (*p* < 0.01) average points score in the assessment of B-scan quality (3.9 ± 0.65 points) and in the evaluation of clinical significance (4.03 ± 0.73 points), compared to the other devices. Conclusions: HHUS systems are able to reliably answer various clinical intensive care questions and are—while bearing their limitations in mind—an acceptable alternative to conventional HEUS devices. Irrespective of this, the present study was able to demonstrate relevant differences in the B-scan quality of HHUS devices from different manufacturers.

## 1. Introduction

Although medical ultrasound has been an integral part of clinical diagnostics for more than 50 years, the last 10 years, in particular, have seen rapid technical developments in both hardware and software. Improvements in the conversion of acoustic ultrasound waves as part of optimised post-processing have led to a reduction in the size and further development of available sonography devices [1,2,3,4]. The trend towards smaller and more compact US devices has culminated in the increasing use of ultra-compact and comparatively inexpensive, pocket-sized devices [5], the significance of which cannot yet be definitively assessed from a qualitative perspective, and which has become the subject of current research [6,7].

The first compact, battery-operated portable US device for bedside use was presented and clinically evaluated as early as 1978—the “cardioscope” [8,9,10]—but it was never used clinically on a large scale. Since the early 2000s, several studies have been conducted on compact, battery-operated portable sonography devices [11,12,13,14]. Under various clinical conditions (vascular, cardiac, and intensive care), this generation of devices has been certified to have acceptable clinical informative value—with the HEUS system used in parallel, serving as a comparison in each case.

A distinction can be made between these compact mobile ultrasound devices with a permanently integrated monitor and ultra-compact handheld ultrasound (HHUS) devices [5,15], which are not significantly larger than a conventional ultrasound transducer, including the battery and electronics. They can be connected to various end devices (tablet, mobile phone, etc.), and for the first time can fit “in your pocket” [15]. Such HHUS devices have been available for around 10 years, and have been evaluated under a variety of clinical conditions [7,16,17,18].

HHUS is becoming increasingly important for intensive care patients, due to its easy application, independent of time and place [19]. The use under intensive care conditions became even more crucial during the coronavirus pandemic, as the small devices were easy to use on isolated patients and proved effective in detecting pulmonary issues [20,21]. It is expected that HHUS will continue to be used more frequently and for several clinical issues in the ICU [22,23].

There are now numerous different manufacturers of HHUS devices [24]. The various devices have been compared in terms of their technical features [24,25], but there are almost no comparative studies on B-image quality and clinical significance. In this regard, a first publication has recently appeared which, in addition to technical aspects, also assesses the B-image quality of four different HHUS devices [26].

The primary objective of the present study is to compare the imaging quality in the B-image between a total of eight current HHUS models from different manufacturers with respect to five previously defined intensive care questions.

In a second step, it will be investigated whether any differences between the tested HHUS devices affect the ability to address the clinical issue.

## 2. Materials and Methods

### 2.1. Study Design

This prospective clinical non-inferiority study compares 8 HHUS devices from different manufacturers under real-life clinical circumstances, with regard to their B-image quality for use in the intensive care setting. A simultaneously used HEUS system serves as a reference, in accordance with previous studies [27,28]

The primary endpoint of the study is defined as the rating of B-image quality, using a 5-point Likert scale (5 points = very good; 1 point = insufficient). Secondary endpoints relate to the ability to address previously defined clinical questions. The 5-point Likert scale was also used for this purpose. The assessment was carried out by several anaesthesia and intensive care medicine specialists who possess a high level of expertise in ultrasound.

Referring to the number E-01-20220502, the study was approved by the ethics committee of the Medical University Theodor Fontane Brandenburg. Written agreement was given by all participating patients, including the usage of all sonographic findings, in accordance with the study protocol. 

Ultra-compact portable sonography devices were used, which can be operated with just one hand, due to their size and weight. These devices contain both a conventional B-image ultrasound and a duplex function, and are summarised under the term HHUS [7,17]. They consist of a convex or combination transducer, which contains the battery and all the electronics for processing the data. The generated ultrasound images are transmitted via cable or wireless to a freely selectable monitor (tablet or mobile phone).

### 2.2. Study Schedule

The study schedule and the applied methodology are shown schematically in Figure 1, in accordance with [27,28]

From July 2022 to November 2022, a total of 8 HHUS devices from various manufacturers were employed in clinical settings. We defined 5 typical sonographic questions which often occur under intensive care conditions.

The first step was to record findings obtained under the previously defined sonographic question in randomly selected intensive care patients, using a traditional HEUS system with stand-alone devices (Canon Aplio i900, Canon Medical Systems Corporation, Ōtawara, Japan); these findings were defined as the highest benchmark. Immediately afterwards, the same findings were recorded with one of the HHUS devices to be tested. Both examinations were performed by the same examiner, and the documentation was undertaken with both the HHUS devices and the HEUS system as a 10s long video clip. Thus, a total of 40 video clips were created, using 5 different sonographic sections and 8 different HHUS, with the participation of 40 test subjects.

The resulting video clips were anonymised in the same way, with regard to patient and manufacturer information, as part of professional digital post-processing, without making digital changes to the actual B-image. Finally, the corresponding clips were displayed side by side, on a screen.

The evaluation was carried out afterwards as a visual comparison, by experienced ultrasound examiners. In an online interview, both ultrasound clips were played simultaneously (for an example, see Figure 2).

The quality of the HHUS device B-image was primarily evaluated using the parallel HEUS-system ultrasound clip of the same pathology, as a comparative standard. Possible quality criteria included good spatial resolution, the grey-scale contrast, and the overall image quality The evaluation was performed using a 5-point Likert scale (very good—good—acceptable—sufficient—insufficient). In a second step, the HHUS video was reassessed with regard to how well the specified clinical question could be answered (the same 5-point Likert scale). For this purpose, the online interview displayed the specific clinical question, along with the assessment video. All answers could be entered digitally by clicking on an appropriate selection box, and the cumulative data were finally sent online for evaluation.

All 8 HHUS devices used were tested in this way, based on the 5 previously defined sonographic questions, in comparison to the high-end device, in terms of B-image quality and then in terms of clinical significance, as also applied to preliminary studies [27].

### 2.3. Technical Resources—Ultrasound Devices

Prior to the study, a market analysis of the German medical device market identified 11 manufacturers (Table 1). Based on this, all manufacturers were asked to provide a HHUS device as part of the study. As a result, 8 devices from different manufacturers were included in the study. All of them used Tissue Harmonic Imaging (THI) for B-image optimization, had a duplex module, and offered different options for data storage (DICOM) on the hard disk or in a corresponding cloud. Two devices were compatible with an Android operating system, and the others could install device-specific software on Windows, Android, and iOS tablets. Some devices had one or two programmable control buttons (e.g., for the “freeze” function), while others could only be controlled through the screen. Due to the focus of the study on the evaluation of B-image visualization, detailed discussions of technical or software differences between individual handheld ultrasound (HHUS) devices are not provided here [6,24].

### 2.4. Human Resources—Patients and Examiners

The pathologies were gathered from patients who required sonography as a component of their inpatient care. A total of 40 patients participated in the study. They were, on average, 76.7 years old (95% confidence interval 72.6–80.7) and had a BMI of 25.03 kg/sqm (95% CI 24.1–25.3). A total of 23 of the patients were male, and 17 were female.

All ultrasound examinations were performed by two highly experienced Level 2 or 3 DEGUM (German Society for Ultrasound in Medicine) examiners. Both of them had performed more than 6000 ultrasound examinations. The study followed a highly standardized protocol. To minimize bias, efforts were made to equalize the initial level of the investigators, in terms of prior knowledge. Prior to using the various HHUS devices, both examiners thoroughly familiarized themselves with the device-specific application modes and image optimization options. It is worth noting that both examiners had only sporadically used HHUS devices prior to the study, and had no specific or detailed prior experience with any of the devices.

### 2.5. The Sonographic Questions

The sonographic questions used are intended to reflect common clinical concerns in intensive care medicine, and are based on guidelines provided by national and international professional societies [29]. Table 2 shows the sectional planes used in the study, including transducer positions and sonographic questions.

The majority of the selected questions involved interfaces with large impedance jumps. Both sections with low penetration depth (needle tracking in situ, pleural effusion) and those with deep penetration depth (volume status/IVC, B-lines) were selected. Depending on the patient’s constitution, the gallbladder was mainly visualised at a medium distance from the transducer. As an example, Figure 3 shows the visualisation of the B-lines of 8 different HHUS devices and a reference image recorded with a HEUS system.

### 2.6. The Raters

The assessment was conducted by consultants who were highly proficient in clinical ultrasound, each having engaged in daily sonography practice for a minimum of 2 years. The examiners were chosen through an existing network of anaesthesiologists experienced in sonography, and they were contacted via telephone or email. Participation was voluntary, and 16 out of 18 doctors who were asked took part in the study, resulting in a high response rate of 89%. A total of 14 of the evaluators are specialists in anaesthesia and intensive care medicine, and 2 are specialists in internal medicine; 5 also have the additional qualification of intensive care medicine. The evaluators work in 5 hospitals in Germany and Austria, 4 of which are university hospitals.

### 2.7. Statistics

Data were collected via Microsoft Excel^®^ version 16.48 (Microsoft Corporation, Redmond, WA, USA). SPSS software version 22.0 (IBM SPSS Statistics^®^, New York, NY, USA) was used to analyse the data.

The individual HHUS devices were evaluated on a five-point Likert scale, by awarding points (5 points for “very good”, 4 points for “good”, 3 points for “acceptable”, 2 points for “sufficient”, and 1 point for “insufficient”). From these obtained scores, it was possible to calculate mean values both for the overall result and for the subgroup analysis, enabling the evaluators’ assessments to be analysed numerically. 

Binary and categorical baseline variables are given as absolute numbers and percentages. Continuous data are given as mean and standard deviation (SD). Categorical variables were compared using Fisher’s exact test, and continuous variables using *t*-test or the Wilcoxon and the Mann–Whitney test. In addition, parametric (ANOVA) or non-parametric (Kruskall–Wallis) analyses of variance were calculated and further explored with pairwise post hoc tests (*t*-test or Mann–Whitney U). All significance tests were performed bilaterally. *p*-values < 0.05 were considered statistically significant.

## 3. Results

### 3.1. B-Scan Quality

The B-scan quality assessment results are presented in Table 3 and Appendix A. 

The HHUS devices were rated between 2.36 ± 1.03 (device H) and 3.9 ± 0.65 (device A), in regard to B-scan quality. Device A was rated the highest among all tested devices, and the difference between this device and the others (devices B-H) reached a significant level (refer to Figure 4a).

For device A, the B-scan quality was considered “acceptable” or better in 93.7% of the reviews. It is the only device that did not receive a rating of “insufficient”. The lowest rated HHUS device (device H) was rated “acceptable” or better in only 50% of the reviews, and was rated “insufficient” in 22% of all reviews (Figure 5a). When assessing B-scan quality based on sonographic sections, the representations of volume status/IVC achieved higher scores than the other four questions (see Figure 4b). 

### 3.2. Clinical Significance

Full results of the evaluated clinical significance analysis are presented in Table 3 and Appendix A.

The HHUS devices were rated between 2.61 ± 1.23 (device H) and 4.03 ± 0.73 (device A), in regard to clinical significance. Here, too, device A was rated the highest among all tested devices, and the difference between this device and the others (devices B–H) also reached a significant level (refer to Figure 6a).

For the highest scored device (device A), the clinical significance was rated “acceptable” or better in 97.5% of the submitted reviews; more than 83% of the given reviews were “good” or “very good”. This was the only device that did not receive an “insufficient” rating (Figure 5b).

The lowest rated HHUS device (device H) was rated “acceptable” or better in only 50% of the reviews, and was rated “insufficient” in 22% of all reviews (Figure 5b). When assessing clinical significance based on sonographic questions, the representations of volume status/IVC achieved higher scores than the other four questions (see Figure 6b).

### 3.3. Comparison of B-Scan Quality and Clinical Significance

As anticipated, the scores obtained for B-scan quality were found to be correlated with the scores for clinical significance. In general, the ability to address clinical inquiries was rated slightly higher than the evaluation of B-scan quality (refer to Appendix A).

### 3.4. Designation of Manufacturers

Device A scored highest in both B-scan quality and clinical significance. It is Vscan Air from General Electric, Boston, MA, USA. Device B was the second best performing device in terms of B-scan quality, as well as clinical significance. It is Clarius C3, Clarius Mobile Health Corp., Vancouver, BC, Canada.

## 4. Discussion

This study compares eight HHUS devices from different manufacturers in terms of B-image quality and clinical significance. It was possible to demonstrate that there are clear differences in B-image quality between the individual devices, and that these differences are probably relevant for clinical application.

### 4.1. General Aspects

The use of HHUS devices is currently reserved primarily for situations in which examination using a conventional HEUS system cannot be easily performed or is not an option [4]. For example, HHUS devices have been extensively tested in emergency situations [21,24,30], in the outpatient clinic [31], and on isolated patients [32,33,34]. Their use in the ICU is also a matter of course [22,23], because patients cannot usually simply be transported from there to a separate US examination room.

For obvious reasons, it cannot be claimed that HHUS devices can be used as fully-fledged HEUS systems [6,7]. While there are extensive assessments of HHUS devices in comparison to HEUS systems [18], the present study is focused on the inter-system variability of several devices of the same quality class [5], namely on the comparison of the B-image sonography of the respective current HHUS models, from a total of eight different manufacturers.

The measurement of inter-system variability of sonography devices of the same quality class has, so far, only been conducted in accordance with very specific issues. There are comparisons of different endosonography devices [35], the CEUS quality of two devices in IBD patients has been compared [36], and there are multiple comparisons of the measurement of shear wave elastography of different devices [37,38]. The ergonomics of different mid-range sonography devices have been compared [39] as has the use of different HHUS devices in the measurement of the optic nerve [4]. Our own working group has already performed comparative studies of the B-image quality of high-end and mid-range sonography devices in the field of internal medicine [28,40].

Only one recently published study has so far been available [26] for comparing the B-image quality of different HHUS devices with respect to emergency or intensive care issues. This study compared four different HHUS devices, among other things, in terms of B-image quality in three healthy subjects. One of the four tested devices clearly came in behind the other three devices, in terms of B-image quality. This result is partly correlated with the results of our investigation. A direct comparison of this study with our investigations is not possible, because we—unlike Le’s working group—collected pathological findings on real patients. Finally, we included twice as many HHUS devices in our study, and focused on the B-image quality of the individual devices with respect to five different clinical questions.

### 4.2. The Sonographic Questions

Our investigation focuses on the application of HHUS devices with respect to intensive care medicine issues, which can be assessed using the B-image. In addition to the detection of fluids (pleural effusion, ascites, abscess, etc.), this includes the assessment of the volume status (diameter and respiratory variability of the IVC, pleural effusions, pleural oedema), and the sonographically supported placement of catheters and drains, along with general internal issues such as the search for an infection site, urinary retention, etc. [41,42,43]. In contrast to several previous studies [26,44,45,46], when selecting our clinical questions we attempted to cover the entire range of B-image ultrasound used in intensive care medicine. To that end, we selected sections with great penetration depth (pleura, IVC) and those with low penetration depth (needle tracking, B-lines). Examination of the gallbladder is given special attention. Although the visualisation of sludge or an oedematous gallbladder wall, for example, is not an everyday intensive care issue, it is precisely in this case—more than with the other selected sections—that the separation of grey tones is required, whereby a HUUS device may be pushed to its technical limits, rather than visualising structures with high-impedance jumps [47,48].

### 4.3. Selection of HHUS Devices

When selecting the HHUS devices within this study, an attempt was made to cover the entire range of HHUS devices available on the German medical device market. Of the 11 HHUS device brands we identified, eight companies provided us with their current HHUS models. This enabled us to access the vast majority of the companies mentioned in previous studies, and thus to test the most representative selection of available HHUS devices.

The proper functioning of the device significantly affects the display in the B-scan [49]. All HHUS devices used in this study had multiple programmed presets for the B-scan, depending on the intended application. Adjusting the B-scan on most devices requires navigating through various submenus within the respective application. The video clips were recorded by two highly experienced examiners who have extensive knowledge in B-scan optimization and are well-versed in the HHUS and its application software. This ensured optimal setup for B-scan sonography. The examiners have used various HHUS devices in the past, but are not familiar with the use of one specific HHUS device over another. So, it is improbable that their familiarity with a device or specific user software influenced them.

The HHUS devices we used differed in terms of software, size, weight, battery life and number of available transducer variants, and in terms of price, available subscription models, and much more. Our investigation focuses explicitly on the B-image quality. Regarding the other differences, we refer to two previous review articles that deal in detail with the technical differences of several available HHUS devices [6,24].

The real axial and horizontal resolution of sonographic images is determined by the spatial arrangement of the sound crystals, the sound frequency used, the post-processing techniques employed, and more [50,51]. Both HHUS and HEUS produce digital images with a minimum resolution of 800 × 600 pixels after post-processing. Although there may be measurable differences in the quality of displays used for HHUS [52], the influence of display resolution on B-scan quality should be negligible, due to the considerably lower real spatial resolution of sonography devices [53,54].

The price of an individual HHUS device may not be directly correlated with the value of the device. Various factors can influence the cost, including market presence, service quality, market policies, and more. The tested devices were priced between USD 2000 and USD 10,000. Due to highly variable pricing models, such as subscriptions, additional cloud usage, user modules, and other online tools, direct cost comparisons are difficult. The two highest-rated devices are priced at around USD 5000 in Germany, placing them in the mid-price range. For more detailed information on the prices of several HHUS devices, please refer to the previous publication by Malik [24].

### 4.4. Study Results

Overall, the HHUS devices were found to have a high level of reliability, both in terms of B-scan quality and clinical significance. Fewer than 10% of the evaluations were rated as “insufficient” (see Figure 5a,b). In the assessment of clinical informative value, the two best-evaluated devices (Device A and Device B) were not rated “insufficient” in any scenario. This overall good performance of the HHUS devices with respect to clinical issues correlates with several previous studies [44,45,55,56,57].

One of the most important results of this study is the evidence of clear inter-system variability of the included HHUS devices. There are obvious qualitative differences in the B-image quality of the various HHUS devices (Figure 3). These differences have a relevant influence on the informative value of the clinical application of HHUS devices. For the worst rated device (Device H), “insufficient” was the score in more than 20% of all submitted evaluations (see Figure 5a,b); in no case was “very good” awarded for the B-image quality of this device. In contrast, the two best devices were in no case rated as “insufficient” in the assessment of clinical informative value, while almost a quarter of the evaluations submitted achieved the rating of “very good”. This can be seen as evidence that the qualitative differences between the tested HHUS devices are not only statistically significant, but also relevant to everyday clinical practice. If a device receives an insufficiency rating in more of 20% of evaluations, it cannot be recommended for safe clinical use. There is a risk of misdiagnosis, which can have fatal consequences for the patient and lead to ethical, financial, and juridical consequences for the user. On the other hand, devices A and B were never rated “insufficient” in terms of clinical relevance. Consequently, these two devices can be fully recommended for use in the intensive care unit, considering the limitations of the present study listed below.

Of the HHUS devices tested, one device scored significantly better than the other devices (Device A: GE Vscan Air), both in terms of B-image quality and clinical significance. In the most comprehensive meta-analysis to date, with 16 publications on the quality of HHUS devices [18], this device was used 15 times, which indicates a high market presence.

As expected, the evaluations of B-image quality and clinical informative value (Appendix A) correlate well with each other, with clinical informative value consistently tending to receive better evaluations than image quality.

The subgroup analysis according to the used ultrasound sectional planes showed a significantly better evaluation of the HHUS devices regarding the assessment of the volume status by visualization of the IVC than the other four clinical questions. (Figure 4b and Figure 6b). This result is unexpected, because the US waves have to penetrate several connective tissue septa, fat layers, and often also the stomach or intestine from the epigastric region, in order to visualise the IVC, which is remote from the transducer; we would have thought that this would be more challenging for the US technology than, for example, visualising a needle in the pleural effusion. As expected, the clinical questions relating to the gallbladder are the most difficult to answer, and this difference was even shown to be significant. As already mentioned above, the separation of grey tones is required more than with the other selected sections for clinical questions with the gallbladder (such as wall thickening, wall oedema, sludge, and concretions). This may push the HHUS devices to their technical limits more than visualising structures with high-impedance jumps [47,48]. This result correlates with experiences from previous studies, in which structures with high-impedance jumps, in particular, were well visualised by HHUS devices [44,45,55,58].

### 4.5. Limitations

An important limitation of the present study is that the US sections of the eight HHUS devices used have not been recorded on the same patient. Unfortunately, it was not possible to test all eight HHUS devices at the same time. As a result, we had to make do by having the findings recorded in parallel with a HEUS, which then served as the evaluation standard. Performing head-to-head comparisons of all eight HHUS devices on the same clinical findings would enable an even more objective assessment.

While similar presets were applied to all handheld ultrasound (HHUS) devices, a potential limitation lies in the varying manufacturer-specific software of these devices. The two highly experienced examiners in this study made a concerted effort to achieve adequate image optimization on all HHUS devices used. The user-friendliness and portability of the HHUS devices were not the focus of our investigation, and so were not evaluated. It would be possible to conceive an influence on the quality of the recorded video sequences.

A bias on the part of the examiners via habituation to a previously used device cannot be completely ruled out. In any case, the two examiners had previously only sporadically used HHUS devices from different manufacturers, and were not used to a specific manufacturer.

Despite a detailed market analysis, it is possible that not all currently available HHUS devices could be requested for the study. In addition, the constant technical development and software updates should also be taken into account, and may affect the informative value of our study. The present study considers only some of the possible applications of HHUS devices under intensive care conditions. B-image ultrasound certainly covers an important and comprehensive part of intensive care ultrasound, but we have not considered echocardiographic issues in our study. These issues and the application of duplex sonography could be the subject of future investigations.

### 4.6. Outlook

Artificial intelligence (AI) is also playing an increasing role in the field of sonography. There are modules for the better sonographic characterisation of malignant lesions in the liver, thyroid and breast [59,60,61], and for the assessment of liver fibrosis [62]. The combination of HHUS devices and AI is described in a first publication on breast tumour screening [61]. A proof-of-concept study on using HHUS devices and AI in abdominal ultrasound was initiated by our own working group [63]. Finally, a paper recently appeared, in which five HHUS devices are used for AI-assisted measurement of the optic nerve sheath [64]. It is to be expected that AI will positively influence both the diagnostic options for the use of HHUS devices and their B-image quality [65,66]. Due to their increasingly acceptable image quality, their compactness, and, not least, their affordable prices, HHUS devices will inevitably be further developed and established within AI-based sonographic diagnostics, and extensively used within telemedical networking.

## Figures and Tables

**Figure 1 diagnostics-14-00054-f001:**
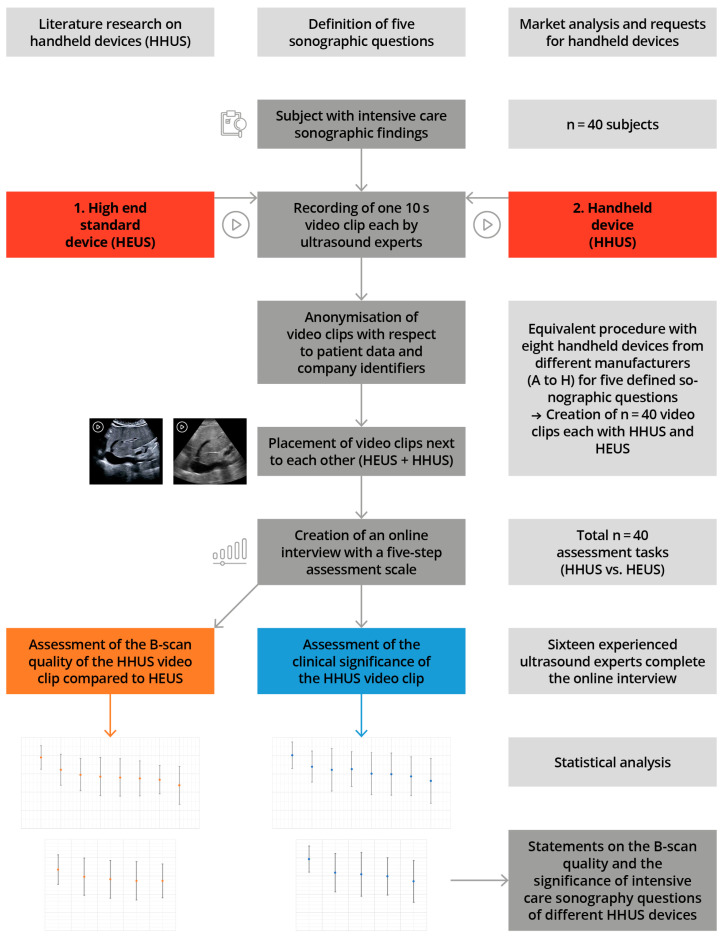
Illustration of the study design, including measurement time points, in accordance with [27,28].

**Figure 2 diagnostics-14-00054-f002:**
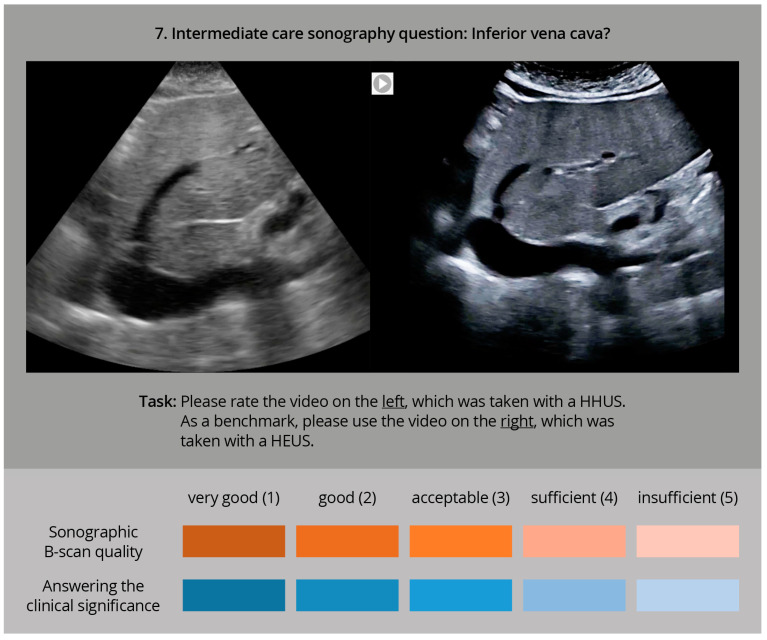
Sample question of the online interview with two sonography videos to be played in parallel (HHUS and HEUS), to be evaluated by the participating ultrasound experts using two rating scales with regard to sonographic image quality and clinical significance, in accordance with [27,28].

**Figure 3 diagnostics-14-00054-f003:**
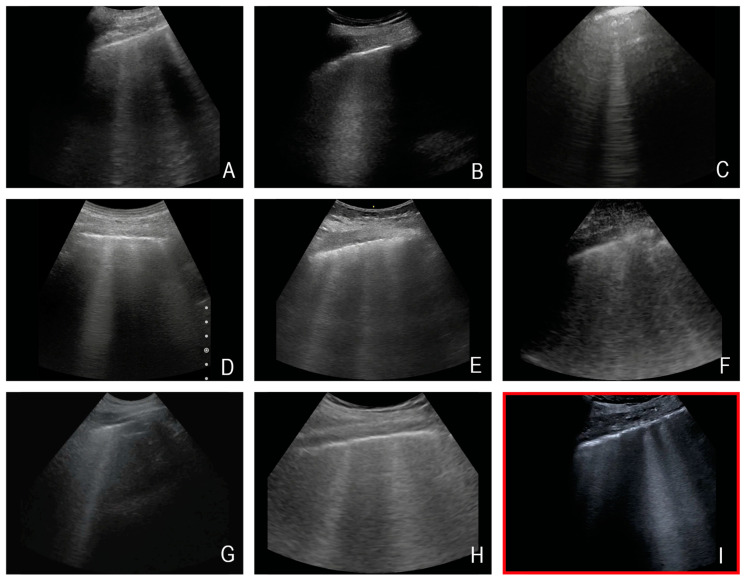
Exemplary representation of the sonographic question “B-lines” by eight different HHUS devices (in the first row, devices (**A**–**C**), in the second row, devices (**D**–**F**), and in the third-row, devices (**G**,**H**)), and the HEUS device (Canon Aplio 900i) used as reference ((**I**)—marked in red). Note: All images are from different subjects. This figure is intended to illustrate an approximate impression of the different representation of B-lines by the various HHUS devices.

**Figure 4 diagnostics-14-00054-f004:**
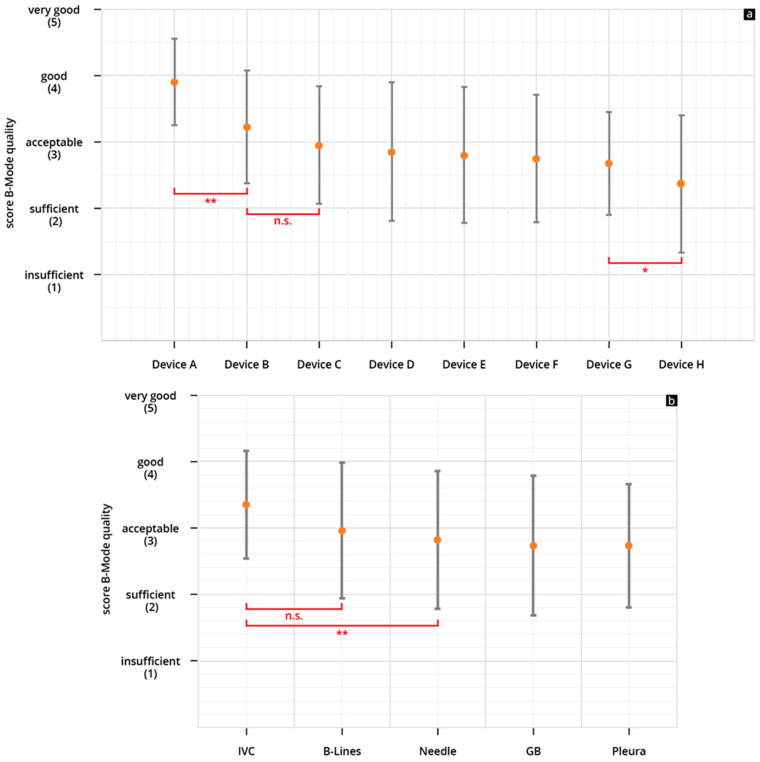
(**a**,**b**) Mean point score and SD achieved in the assessment of the B-scan quality of the individual HHUS devices (**a**) and mean cumulative scores for all HHUS devices and SD as a function of the sonographic sections (**b**) (* *p* < 0.05; ** *p* < 0.01; n.s., not significant; Analysis of Variance (ANOVA analysis). The order in which the individual HHUS devices were displayed was determined by the score achieved. The device with the highest point score in the evaluation of B-scan quality is referred to as Device A; the following devices are accordingly designated Device B, C, etc., and the device with the worst rating, Device H. A complete list of the *p* values can be found in the Appendix A.

**Figure 5 diagnostics-14-00054-f005:**
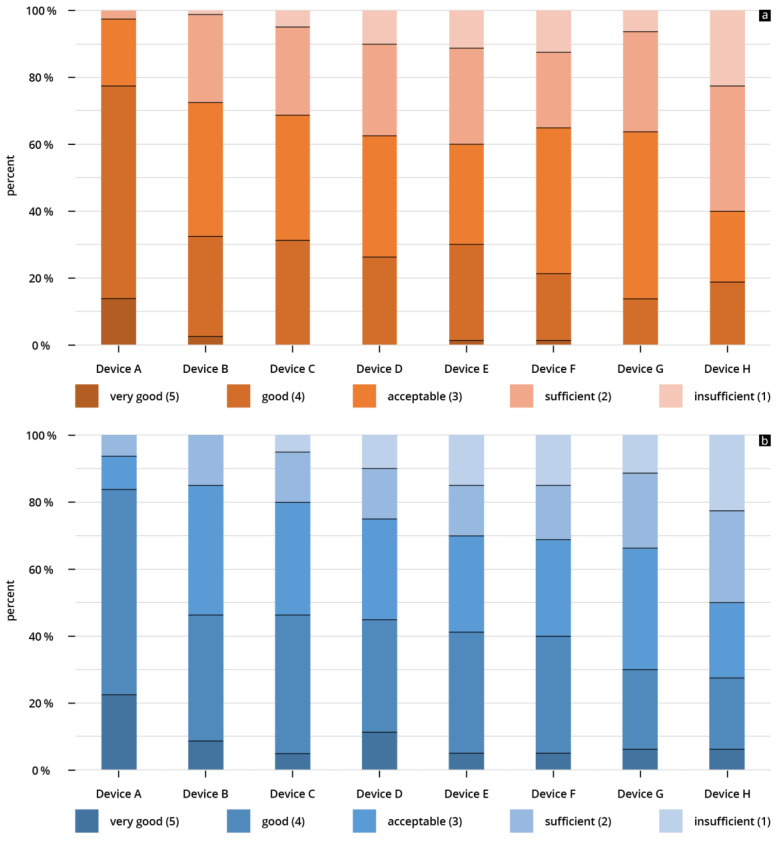
(**a**,**b**) Percentage of given ratings (cumulative) for (**a**) B-scan quality of the individual HHUS devices and (**b**) clinical significance of the individual HHUS devices. A complete list of the *p* values can be found in the Appendix A.

**Figure 6 diagnostics-14-00054-f006:**
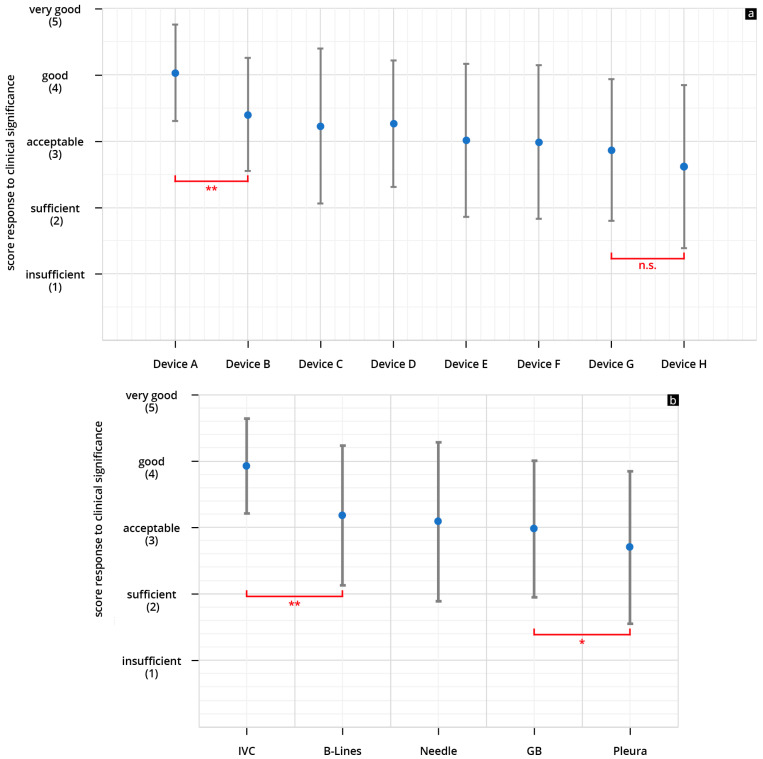
(**a**,**b**) Mean point score and SD achieved in the assessment of the ability of the individual HHUS devices to answer the clinical question (**a**) and mean cumulative scores for all HHUS and for SD as a function of the sonographic sections (**b**) (* *p* < 0.05; ** *p* < 0.01; n.s., not significant). The order in which the individual devices were displayed was as shown in Figure 4a, depending on the score achieved in the evaluation of the B-scan quality. A complete list of the *p* values can be found in the Appendix A (GB = Gallbladder).

**Table 1 diagnostics-14-00054-t001:** Presentation of the HHUS devices requested or included in the study (in alphabetical order). * these devices could not be included, as either no feedback was received from the manufacturer or device delivery was not possible.

Device Name	Manufacturer	Country	Image Transmission	Transducer
Alpinion minisono	Alpinion Medical Systems	Seoul, Republic of Korea	Wired	Convex or linear
Butterfly iQ+	Butterfly Network, Inc.	Burlington (MA), USA	Wired	“1.75 D-Array”
Clarius C3 HD3	Clarius Mobile Health Corp.	Burnaby (BC), Canada	Cordless	Multiple attachable probes
iSiniQ 30A *	Prunus Medical Shenzhen	Shenzhen, China	Cordless	
Kosmos	EchoNous Inc.	Redmond (WA) USA	Wired	All-in-one transducer
mSonics MU 1 *	Lang Sheng Sozhou	Guangdong, China	Unknown	
Philips Lumify	Koninklijke Philips N.V.	Amsterdam, The Netherlands	Wired	Broadband convex and two other variants
SonoSite iViz	FUJIFILM Corporation	Tokio, Japan	Wired	All-in-one transducer
Sonostar * Uprobe-C4PL	Universal Diagnostic Solutions	Vista (CA), USA	Cordless	Convex and linear in one
Vscan Air	General Electric	Boston (MA) USA	Cordless	Convex and linear in one
Youkey Q7	Wuhan Youkey Bio-Medical Electronics Co., Ltd.	Wuhan, China	Cordless	Replaceable transducers

**Table 2 diagnostics-14-00054-t002:** Characterisation of the five sonographic questions used in relation to the organ, sonographic question, section, sonographic field, and penetration depth.

No.	Organ	Sonographic Question	Sonographic Section	Near vs. Far Field	Penetration Depth
1.	Inferior Vena Cava (IVC)	Diameter and variability of IVC, volume status	Median sagital section	Far field	up to 12 cm
2.	Pleural effusion	Presence of effusion	Flank section	Far field	up to 12 cm
3.	Needletracking	Visibility of the needle	Variable: abdomen or intercostal	Near field	up to 6 cm
4.	B-Lines	Presence and number	Thoracic, intercostal	Near field	up to 4 cm
5.	Gallbladder	Wall thickening, sludge, oedema, stones	Diagonal section—right upper abdomen	Near/middle field	up to 8 cm

**Table 3 diagnostics-14-00054-t003:** All results depend on the clinical questions and the devices used.

B-Mode Quality						
Device	IVC	Needle	B-Lines	GB	Pleura	Cumulative Score
	mean ± SD	mean ± SD	mean ± SD	mean ± SD	mean ± SD	mean ± SD
Device A	4.25 ± 0.44	3.43 ± 0.63	4.06 ± 0.57	4.00 ± 0.52	3.75 ± 0.77	3.90 ± 0.65
Device B	2.69 ± 0.71	3.69 ± 0.79	2.87 ± 0.88	2.94 ± 0.85	3.12 ± 0.72	3.22 ± 0.85
Device C	3.44 ± 0.73	2.00 ± 0.73	3.81 ± 0.40	2.50 ± 0.52	3.00 ± 0.63	2.95 ± 0.88
Device D	3.94 ± 0.85	2.36 ± 0.81	1.88 ± 0.72	3.25 ± 0.93	2.81 ± 0.54	2.85 ± 1.04
Device E	2.94 ± 0.68	3.81 ± 0.54	3.44 ± 0.63	1.56 ± 0.51	2.25 ± 0.77	2.80 ± 1.02
Device F	3.06 ± 0.57	3.18 ± 0.66	2.13 ± 0.81	3.56 ± 0.73	1.81 ± 0.75	2.75 ± 0.92
Device G	3.00 ± 0.52	2.50 ± 0.89	2.25 ± 0.77	2.44 ± 0.51	3.19 ± 0.75	2.67 ± 0.78
Device H	3.50 ± 0.63	1.56 ± 0.51	3.25 ± 0.68	1.62 ± 0.72	1.87 ± 0.50	2.36 ± 1.03
Clin. Signif.						
Device	IVC	Needle	B-lines	GB	Pleura	Cumulative Score
	mean ± SD	mean ± SD	mean ± SD	mean ± SD	mean ± SD	mean ± SD
Device A	4.37 ± 0.62	3.50 ± 0.62	4.31 ± 0.48	4.06 ± 0.57	3.87 ± 0.72	4.03 ± 0.73
Device B	3.56 ± 0.63	3.94 ± 0.85	3.00 ± 0.82	3.00 ± 0.89	3.50 ± 0.73	3.40 ± 0.85
Device C	3.75 ± 0.58	2.44 ± 1.09	4.06 ± 0.44	2.63 ± 0.62	3.44 ± 0.73	3.26 ± 0.95
Device D	4.44 ± 0.73	3.00 ± 1.26	2.13 ± 0.80	3.13 ± 1.02	3.44 ± 0.63	3.23 ± 1.17
Device E	3.44 ± 0.73	4.06 ± 0.57	3.56 ± 0.63	1.44 ± 0.51	2.56 ± 0.96	3.01 ± 1.15
Device F	3.63 ± 0.50	3.75 ± 0.86	2.25 ± 1.06	3.56 ± 0.81	1.75 ± 0.77	2.98 ± 1.15
Device G	4.12 ± 0.62	2.00 ± 0.89	2.50 ± 0.73	2.37 ± 0.81	3.31 ± 0.70	2.86 ± 1.06
Device H	4.06 ± 0.68	2.00 ± 0.73	3.62 ± 0.62	1.37 ± 0.50	2.00 ± 0.73	2.61 ± 1.23

## Data Availability

The data presented in this study are available on request from the corresponding author. The data are not publicly available, due to institutional and national data policy restrictions imposed by the ethics committee, since the data contain information that could potentially identify study participants. Data are available upon request (contact via weimer@uni-mainz.de) for researchers who meet the criteria for access to confidential data (please provide the manuscript title with your enquiry).

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
