# Peer review of "Inter-System Variability of Eight Different Handheld Ultrasound (HHUS) Devices—A Prospective Comparison of B-Scan Quality and Clinical Significance in Intensive Care"

_diagnostics, 2023, doi:10.3390/diagnostics14010054_

Round 1
Reviewer 1 Report
Comments and Suggestions for Authors
This is a research article entitled "Inter-system-variability of eight different handheld ultrasound devices (HHUS) – a prospective comparison of B-scan quality and clinical significance in intensive care". Ultrasound images obtained from 40 inpatients by using 8 different HHUS devices were compared, including B-mode quality and clinical significance. The authors concluded that HHUS systems can reliably reproduce findings and are an acceptable alternative to conventional HEUS.
There are several points to be concerned in this study.
Methods:
1. Please describe the procedure of ultrasound examination procedures in detail. In Figure 1, HHUS was used before HEUS. While in the text, HEUS was used before HHUS.
2. Did the patient undergo two ultrasound examinations by the same operator (first with the HEUS system and then with the HHUS system), or did the patients receive two more ultrasound examinations (with the HEUS system and then with various HHUS systems)? Although this point is mentioned in the [Limitation] section, clarification is also necessary in the [Methods] section.
3. Please describe the resolution of each ultrasound image, including HEUS and HHUS systems.
4. Based on the methodology, this was a head-to-head comparison between one of the HHUS and the HEUS systems. How many ultrasound examinations were performed on 40 patients? Were all HHUS systems used to study all targe organs listed in Table 2? If not, a summary table showing how often each HHUS system was used in different organs is needed.
5. Please explain the reason for selecting 16 evaluators for scoring.
6. Figure 3 demonstrates the finding of each ultrasound system. Were these images detected from a same patient with the same depth setting? Please also provide the illustrative example of the rater’s reading score for each image’s B-mode quality and clinical significance (perhaps indicated in white text at the bottom of each image. Therefore, it’s better to arrange images in order of B-mode quality or clinical significance.
7. Please reconfirm whether parametric or non-parametric methods were more appropriate for analysis.
8. A brief description of the price range for each device may be help readers compare the quality and cost.
Results:
1. “The HHUS devices used scored between 2.36 ± 1.03 (Device H) and 3.0 ± 0.65 (Device A) in terms of B-scan quality (Table 3).” Did aforementioned means represent the average scores of 5 organs in each device? If so, please add these values to the end of each device in Table 3.
2. The numbers listed in Table S1 are difficult to understand, such as “4,2”, “3,8”.
3. Please describe the statistical methods used for Figure 4 (ANOVA or Kruscal-Wallis?).
4. Figure 4: Please add score next to the Y-axis items, such as very good (5), good (4), etc.
5. The description of Figure 4b is not noted in the text. Were these scores calculated based on an average of 8 devices? A similar question arises in Figure 6b.
6. Section 3.2: “Here, too, the best-rated device (Device A) showed a significantly better point score than the next-best device (De-vice B) (Figure 6).” Should it be Figure 6a?
7. Please provided statistical methods and p values for Figure S1.

Author Response
Dear Reviewer,
thank you very much for your constructive and critical comments! We are submitting a revised version of our manuscript, which we thoroughly updated according to your suggestions. We have attached our responses at the end of this letter, and we have addressed your comments, point-by-point.
Kind regards,
Johannes Weimer on behalf of all authors

Reviewer 2 Report
Comments and Suggestions for Authors
-
- Consider providing a more explicit emphasis on the increasing role of handheld ultrasonography (HHUS) devices within critical care settings, underscoring the relevance of the study within this evolving landscape.
-
- Clarify the specific criteria employed by the sixteen intensive care physicians when utilizing the five-point Likert scale to assess B-scan quality and clinical significance for enhanced understanding.
-
Results:
- Further define or contextualize the term "satisfactory results" to establish a clearer benchmark for acceptable performance across the various HHUS devices.
-
- Expand upon the potential implications arising from the notably superior performance of one HHUS device over others in both B-scan quality and clinical significance. Elaborate on how these discrepancies could influence critical decision-making in clinical settings.
-
- Highlight the practical implications derived from the study's findings regarding the selection of HHUS devices in intensive care environments, reinforcing the relevance of these results for practitioners.
-
- Investigate potential confounding variables that might have influenced the assessment, such as variations in operator familiarity or experience with different HHUS devices.
Author Response

(The authors gave the same response as above.)
